# Actively addressed single pixel full-colour plasmonic display

Daniel Franklin[1,2], Russell Frank[2], Shin-Tson Wu[3] & Debashis Chanda[1,2,3]

Dynamic, colour-changing surfaces have many applications including displays, wearables and active camouflage. Plasmonic nanostructures can fill this role by having the advantages of ultra-small pixels, high reflectivity and post-fabrication tuning through control of the surrounding media. However, previous reports of post-fabrication tuning have yet to cover a full red-green-blue (RGB) colour basis set with a single nanostructure of singular dimensions. Here, we report a method which greatly advances this tuning and demonstrates a liquid crystal-plasmonic system that covers the full RGB colour basis set, only as a function of voltage. This is accomplished through a surface morphology-induced, polarization-dependent plasmonic resonance and a combination of bulk and surface liquid crystal effects that manifest at different voltages. We further demonstrate the system's compatibility with existing LCD technology by integrating it with a commercially available thin-film-transistor array. The imprinted surface interfaces readily with computers to display images as well as video.

[1] Department of Physics, University of Central Florida, 4111 Libra Drive, Physical Sciences Bldg. 430, Orlando, Florida 32816, USA. [2] NanoScience Technology Center, University of Central Florida, 12424 Research Parkway Suite 400, Orlando, Florida 32826, USA. [3] CREOL, The College of Optics and Photonics, University of Central Florida, 4304 Scorpius St, Orlando, Florida 32816, USA. Correspondence and requests for materials should be addressed to D.C. (email: Debashis.Chanda@creol.ucf.edu).

Structural colour arising from plasmonic nanomaterials and surfaces has received ever increasing attention[1–12]. These nanostructures have demonstrated diffraction limited colour through the subwavelength confinement of light and can produce the smallest colour generating elements physically possible. Along with the ability to control phase and polarization, these metallic nanostructures could lead to very small pixels useful for next generation projection or 3D displays. The drive to commercialize these systems has also led to significant improvements in colour quality[13,14], angle independence[12,15], brightness[16] and post-fabrication tunability[17]. But while most of these advances struggle to replace present commercially available technologies, the ability to change colour, post-fabrication, is an advantage of plasmonic systems which may allow it to fill niche applications. For example, traditional transmissive and reflective displays typically have three sub-pixel regions with static red, green and blue colour filters. These sub-pixels control the amount of each basis colour transmitted or absorbed to create arbitrary colours through a process called colour mixing. On the contrary, a display built from a dynamic colour changing surface can eliminate the need for individual sub-pixels, increasing resolution by three times without reducing pixel dimensions. In our previous work, we demonstrated that the range of plasmonic colour tuning could span the visible spectrum by using nanostructures of several periodicities in conjunction with a high birefringent liquid crystal (LC)[17]. However, this and other reports of post-fabrication plasmonic tuning have yet to span an entire colour basis set (red-green-blue (RGB) or CYM) with a single nanostructure[18–22].

Here, we demonstrate a reflective colour changing surface capable of producing the full RGB colour basis set, all as a function of voltage and based on a single nanostructure. This is achieved through a surface morphology-induced, polarization-dependent plasmonic resonance and a combination of interfacial and bulk LC effects. Each of these phenomenon dictate the colour of the surface within different voltage regimes: bulk LC reorientation leading to polarization rotation[23] in the low voltage regime, and surface LC reorientation leading to plasmonic resonance shifting at higher voltages. The hybrid LC-plasmonic tuning mechanism is modelled through a combination of finite element (FEM), Jones calculus and finite difference time domain (FDTD) simulation techniques. Lastly, we demonstrate the scalability and compatibility of this system with existing LCD technology through integration with a thin-film-transistor array (TFT). The resultant device is then interfaced with a computer to display arbitrary images and video. This work demonstrates the potential of hybrid LC-plasmonic systems for single pixel, full-colour high resolution displays and colour changing surfaces.

## Results

**Liquid crystal-plasmonic device.** A schematic of the LC-plasmonic system is shown in Fig. 1. At the top of the device, unpolarized ambient light passes through a linear polarizer, glass superstrate, indium tin oxide (ITO) and a rubbed polyimide film. The ITO serves as a transparent electrode for applying electric fields across the LC, and the rubbed polyimide aligns the LC parallel to the axis it is rubbed (homogeneous alignment). The polarized light continues through a high birefringence LC layer (LCM1107, LC Matter Corp.) and excites grating coupled surface plasmons (GCSP) on the nanostructured aluminium surface. The LC orientation near the plasmonic surface determines the effective refractive index of the GCSP modes and, therefore, the resonant wavelength. Light that is not absorbed by the nanostructure is reflected back from the device resulting in a perceived colour. The plasmonic surface constitutes the second half of the

LC cell and is fabricated through the nanoimprint lithography[24] of a 300 nm period, nanowell array with an impression depth of 80 nm which is followed by a 30 nm-thick aluminium electron beam evaporation. By changing deposition conditions and substrate temperatures, the surface quality and oxide content of the film can be greatly varied[25,26]. We use this to create a relatively rough aluminium surface ($\sim$30 nm grain diameter) which induces a polarization dependence on the GSCP resonance in the presence of an anisotropic media, whereas it was shown in our previous work that a smooth surface ($\sim$10 nm grain diameter) results in near polarization independence[17]. This results in the device having two orthogonal colour states in the 'voltage off' state ($V_{off}$), depending on the polarization of exciting light either parallel (blue) or perpendicular (red) to the LC director near the nanostructure. Also from this previous work, we found the periodic nanowell structure imparts a weak diagonal alignment on the LC with respect to the structure's grating vectors. We use this to create a 45° twisted nematic cell by aligning the top rubbed polyimide layer with one of the grating vectors of the nanostructure. By applying a voltage across the plasmonic film and the top ITO, the orientation of the LC throughout the cell is controlled. At intermediate voltages, the bulk LC reorients and retards the propagating light resulting in a superposition of the surface's two orthogonal 'off-state' modes. At a specific voltage ($V_{th}$), however, the polarization is effectively rotated to excite the opposite plasmonic mode; marking a transition in colour from red to blue, or blue to red. This birefringent effect is wavelength dependent and can add an oscillatory absorption to the reflection spectra as light is partially absorbed on the second pass through the polarizer. However, we will show below that when the polarizer is aligned in parallel with the LC director, the colour of the surface is independent of this effect. At higher voltages, the bulk LC saturates vertically while the LC near the aluminium surface begins to reorient which increases the effective refractive index experienced by the plasmonic nanostructure. This results in a continuously tunable red shift of the GCSP resonance, eventually saturating when the LC near the surface is also completely reoriented by the electric field. In this state ($V_s$), the surface turns green and loses its polarization dependence.

**Polarization-dependent colour.** The colour of the surface originates from absorptive GSCP resonances characterized by[27] $\boldsymbol{k}_{GCSP} = \frac{2\pi}{\lambda_0} \sqrt{\frac{\varepsilon_{AL}\varepsilon_{LC}}{\varepsilon_{LC} + \varepsilon_{LC}}} u + i\frac{2\pi}{P_i}i + j\frac{2\pi}{P_j}j$. Here, $\boldsymbol{k}_{GCSP}$ is the GCSP wavevector, $\lambda_0$ the incident wavelength, $\varepsilon_{LC}$ and $\varepsilon_{AL}$ the dielectric constants of the LC and aluminium, respectively, $\boldsymbol{u}$ is the unit vector of the surface plasmon before bragg scattering, $i$ and $j$ are the mode orders, and $P_i$ and $P_j$ are the structure periodicities along the grating vectors $\mathbf{i}$ and $\mathbf{j}$. The permittivity of aluminium depends greatly on the local morphology of the surface and behaviour can vary drastically with grain size, surface roughness (rms height variation and correlation length) and oxide content[25,28]. An increase in the rms roughness of a metal film causes a red-shift in the resonant plasmon wavelengths and is commonly attributed to an increase in scattering[29]. This red-shift is accompanied by resonance broadening as an increase in scattering also decreases the interaction length of the plasmon resonance—inducing a lower $Q$-factor. While this is normally considered detrimental for plasmonic applications like biomolecular sensing, SERS enhancement and those requiring long plasmon propagation lengths, this effect can be used advantageously for plasmonic structural colour as structures that absorb broad wavelengths of light are able to produce colours not possible from those with only narrow absorption resonances. To demonstrate the impact of rms surface roughness on the

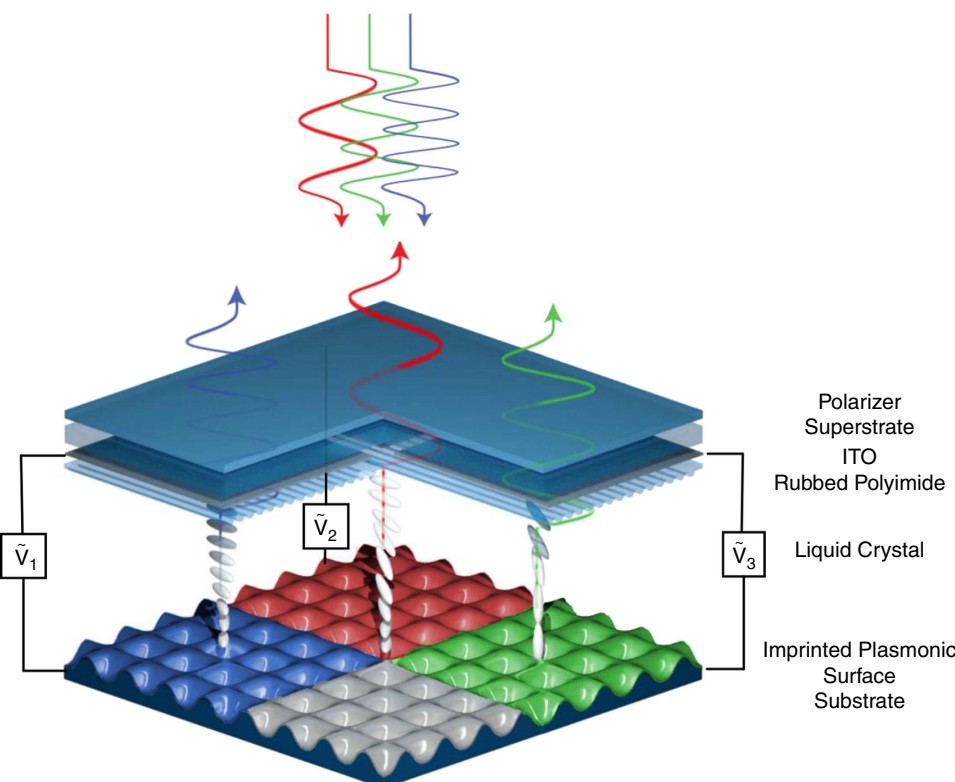

**Figure 1 | Liquid crystal tunable plasmonic device.** Schematic of the liquid crystal-plasmonic device, which works with ambient white light. Light passes through a polarizer, substrate and high-birefringence liquid crystal layer to interact with a continuous aluminium nanostructure. This plasmonic surface exhibits a polarization dependence that originates from surface roughness in the presence of an anisotropic media. The surface absorbs light while reflecting the rest back out of the device. The wavelength of this absorption depends on the liquid crystal orientation near the surface and the polarization of incident light. By applying a field across the cell, the orientation of the bulk LC and interfacial LC can be changed within different regimes, resulting in red, green and blue reflection.

colour reflected from the nanostructure, we perform FDTD simulations of the periodic nanowell array by changing the amplitude of a uniform random height variation about the top surface of the aluminium film while maintaining a constant lateral correlation length of 15 nm. We then use the resulting spectra to predict a reflected colour through the 1932 International Commission on Illumination (CIE) colour space and colour matching functions. Further details about the surface generation and simulations can be found in the Methods section. Here, we approximate the LC region as a perfect anisotropic crystal with the slow axis ($n_e$) parallel to the surface and at 45° with respect to a periodicity vector of the nanowell array (homogeneous LC alignment). In reality, the LC director aligns to the contours of the aluminium surface, resulting in a complex LC director tensor dependent on the local topography of the surface. However, to isolate the effects of rms surface roughness on the GCSP resonance alone, we choose to simplify this LC layer and leave it constant for the purposes of the numerical demonstration in Fig. 2a,b. By separately injecting light polarized parallel and perpendicular to this diagonal, we also isolate the fundamental modes of the plasmonic film and remove any bulk retardation/polarization rotation effects of the LC. The results for incident light polarized perpendicular to the LC long-axis (90°) are shown in Fig. 2a while incident light polarized parallel (0°) to the LC orientation is shown in Fig. 2b. Line colours are determined by the CIE colour matching functions and are cascaded to show the influence of rms surface roughness on resonance location (solid black lines) and full-width-half-maximum (dotted black lines). At low values of surface roughness, the plasmonic resonance shifts less than 20 nm upon incident polarization rotation, resulting in a

minute colour change. However, as the roughness of the aluminium increases, the parallel resonance red shifts more than the perpendicular case, causing a greater colour shift between the two polarization states. We explain this phenomenon with the following argument: the $\varepsilon_{LC}$ term of the above equation can be thought of as an effective index experienced by the plasmonic mode. For isotropic and uniform surrounding media, this term reduces to the dielectric constant of the medium. However, for anisotropic media, this term becomes an overlap integral between the plasmonic field tensor and the surrounding media's dielectric tensor. As plasmonic fields near the metallic interface are normal to the surface, plasmon resonances depend largely on the surface normal component of the surrounding refractive index. Rough films will have local regions where the surface norm has an increased $x$ or $y$ component compared to the global norm. This gives the in-plane components of the surrounding media a greater contribution to the effective refractive index of the GSCP resonance. In the current case, where the surrounding media is a uniaxial crystal oriented parallel to the aluminium surface, this creates a surface roughness induced polarization dependence of the plasmonic resonance.

We experimentally demonstrate this effect with two surface morphologies labelled as A and B that are obtained by controlling deposition parameters and substrate temperatures, detailed descriptions of which can be found in the Methods section. The corresponding top and cross-sectional scanning electron microscope images of the 30 nm aluminium films on the imprinted polymer are shown in Fig. 2c,d. A watershed-based image segmentation method is used to find the distribution of grain

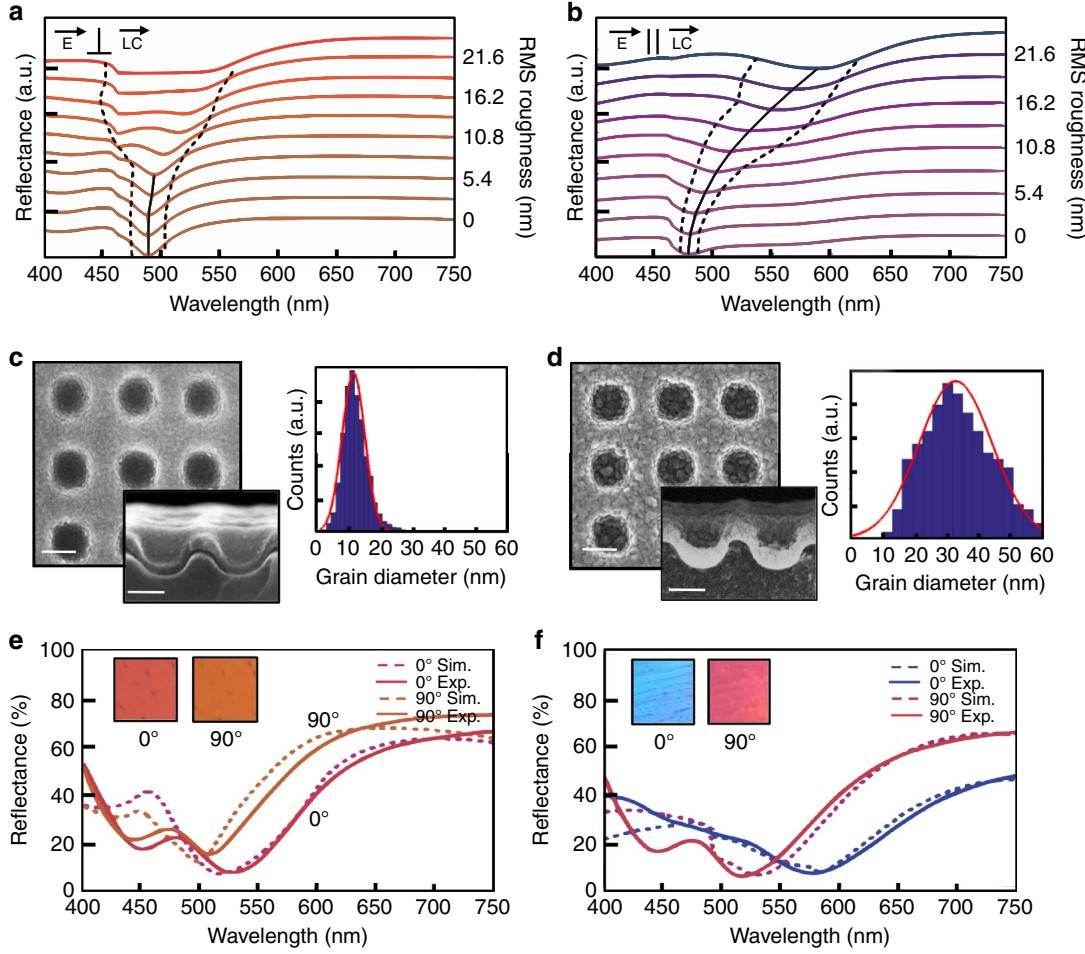

**Figure 2 | Surface morphology induced polarization dependence.** FDTD-computed reflection spectra of a 300 nm period nanowell array with a 30 nm aluminium film for (**a**) incident light polarized perpendicular to the liquid crystal director and (**b**) incident light polarized parallel to the LC director. Line colours are determined by the CIE colour matching functions. Dotted black lines outline the FWHM of the resonance while a solid black line traces the resonance location. (**c**) SEM images of a 'smooth' aluminium film and a corresponding grain size histogram while **d** depicts that of a 'rough' aluminium film. Experimental and simulated reflection spectra of the (**e**) 'smooth' and (**f**) 'rough' surfaces shown above. Scale bars, (**c,d**) 150 nm. Solid lines are FTIR measured reflection spectra while dotted lines are histogram weighted FDTD spectra. Insets show microscope images of the surfaces and all line colours are determined by the CIE colour matching functions.

diameters from the top-view SEM images (Fig. 2c,d). This method is described in detail in Supplementary Note 1. Histograms of the two surfaces show a mean grain diameter of 12 nm for the 'A' surface (Fig. 2c) and 33 nm for the 'B' aluminium surface (Fig. 2d). The surfaces are then made into LC cells and infiltrated with LCM1107. Figure 2e,f shows the resulting reflection spectra of the respective surfaces (solid lines) compared to simulated spectra (dotted lines) for incident polarizations parallel (0°) and perpendicular (90°) to the top rubbed polyimide alignment layer. We show below that in the 0 V state, light polarized parallel or perpendicular to the top rubbed polyimide layer is guided along the twist of the LC as it propagates to the plasmonic surface. Therefore, the experimental polarization excitation (which is with respect to the top polyimide) is equivalent to that of the simulation (which is with respect to the LC director) and we label them 0° and 90°, respectively. Output light of all polarizations is collected to ensure the spectra originate purely from the plasmonic surface. Line colours of the spectra are generated by the CIE colour matching method while the microscope image insets show the experimentally achieved colours. As grain diameter and rms surface roughness are proportional, we use the histograms of the respective aluminium surfaces as a weighted average of the

FDTD spectra of Fig. 2a,b to more accurately represent the surface roughness distribution of the experimental films, which differ from the uniform random distribution of the simulations. The relief depth of the structure in FDTD simulations has also been varied between 80 and 90 nm to provide a better matching to experimental spectra and account for small variances in the imprinting process. The relatively smooth aluminium film, A, gives a small change in colour upon polarization rotation, from an orange to a reddish-orange, associated with a 20 nm shift in GCSP resonance. On the contrary, the aluminium surface, B, with higher surface roughness gives a more drastic colour change upon polarization rotation, from blue to red (Fig. 2f) and is the result of a 60 nm shift in resonant wavelength. Hence, by controlling the surface morphology via deposition conditions, the degree of polarization dependence and the resultant colour can be controlled. Exciting the surface with polarization angles between 0° and 90° or unpolarized light results in a superposition of these two spectra. An example of this is that the surface B when excited with unpolarized light results in a purple (Supplementary Fig. 2). We believe discrepancies between the experimental and numerical spectra of Fig. 2e,f occur for the higher order mode due to the approximation of the LC as a uniform and anisotropic crystal.

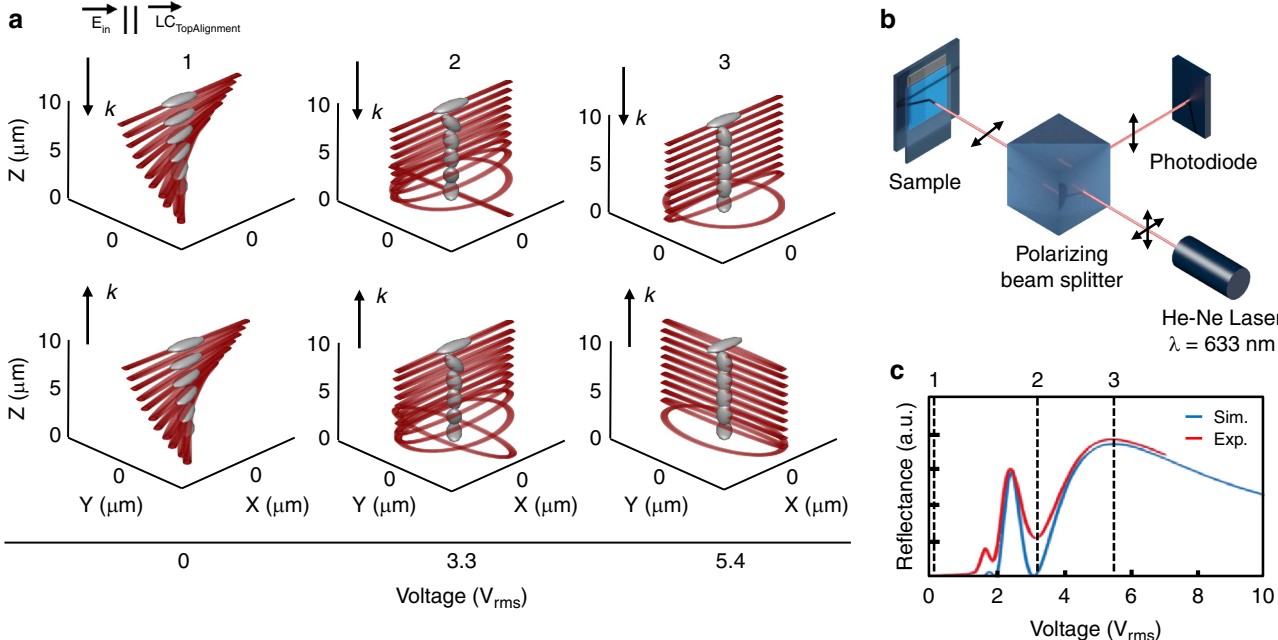

**Figure 3 | Liquid crystal orientation and polarization rotation.** (**a**) Schematics of the liquid crystal and light polarization throughout the cell and at selected voltages. Liquid crystals are not drawn to scale and are down sampled from the FEM simulations. (**b**) Experimental setup to verify liquid crystal orientation throughout cell and (**c**) results. Dotted black lines and labels correspond to the selected fields shown in **a**.

**Liquid crystal phase retardation.** To achieve all three colours of the RGB basis set as a function of voltage we utilize a combination of bulk and surface LC effects. The periodic nanowell structure imparts a weak diagonal alignment on the LC with respect to the structure's grating vectors and we use this to create a 45° twisted nematic cell by aligning the top rubbed polyimide layer with one of the grating vectors of the nanostructure. A cell gap of 8.5 μm ensures the alignment of the top polyimide does not overpower that of the nanostructure to result in a homogeneous cell. These boundary conditions, along with the LC material parameters[17], are used in FEM simulations (TechWiz LCD 3D, Sanayi) to find the LC director throughout the cell as a function of voltage. A Jones matrix approach is then used to find the polarization of light as it propagates through the cell[30,31]. This is done by approximating the LC cell as a stack of $N$ number of phase plates with a continuous variation in retardation. The LC director at that cell location determines the anisotropic refractive index of each layer. A detailed description of the process and examples of the 3D director fields along with the twist and tilt of the LC as a function of cell location and voltage are shown in Supplementary Fig. 3. The resulting LC directors and polarization ellipse throughout the cell for three selected voltages are represented in Fig. 3a. While the FEM and Jones matrix calculations are found to converge with $N = 100$, the figures in Fig. 3a depict a gridded subsampling for graphical purposes. The top row shows the polarization ellipse of light as it propagates down towards the nanostructure, while the bottom row depicts the light after reflection and as it propagates out of the cell. At $V = 0$ (1), the LC creates a 45° twisted nematic cell where the propagating light maintains its linear polarization and is rotated by the LC director in what is known as the Maugin (or Waveguiding) regime. This is given by the following condition: $\phi \ll (2\pi/\lambda)\Delta n d$, where $\phi$ is the twist angle of the cell, $\Delta n$ is the birefringence of the LC, $\lambda$ is the wavelength of light and $d$ is the thickness of the LC cell. Here we can see that light exits the cell with the same polarization as input and is, therefore, unaffected by a second pass through the polarizer. As voltage is increased, the Fredericks transition marks the initial tilt of the LC from their voltage-off state and is followed by a continuous change in tilt along the applied electric field. This tilting reduces the effective $\Delta n$ of the cell, which eventually breaks the Maugin condition and begins changing the ellipticity of the light, as can be seen at points 2 and 3 of Fig. 3a. To verify this LC mode, we use the experimental setup illustrated in Fig. 3b and compare with simulations from the combined FEM-Jones approach. The experiment consists of a He-Ne laser incident upon a polarizing beam splitter and then upon the plasmonic-LC device. In this case, the polarization of incident light is parallel to the rubbing of the top alignment layer. Light is reflected from the device and back into the polarizing beam splitter, which reflects the orthogonal polarization of light to a silicon photodiode. A voltage is then adiabatically applied to the sample at a rate of 0.01 V s$^{-1}$, well below that of the cell's transient optical effects. The results of the experiment are shown in Fig. 3c and represent the degree of polarization rotation imparted on the light as a function of voltage. Jones matrix simulations of a 45° twisted nematic cell with a parallel input polarizer and perpendicular output analyser (both with respect to the top LC director) match well with the experimental curve. A close to zero reflection indicates that the light is leaving the cell with the same polarization as it entered. This can only occur if the light reflects off the plasmonic surface in a linear polarization state, as can be seen in the selected voltage cases A and B. On the contrary, peaks in the curve indicate voltages at which light leaves the cell at a perpendicular polarization than as it entered, which occurs due to the change in hand of reflected circular polarized light (C). While this verifies the bulk LC mechanics of the cell, to understand this effect's impact on the colour of reflected light we look at the polarization of light as it excites the plasmonic surface. The two orthogonal modes of the surface-B are shown in Fig. 4a. These spectra are obtained at 0 V and when the polarization of incident light is parallel (0°) and perpendicular (90°) to the top polyimide alignment layer, respectively. Since the cell is in the Maugin regime at this state, the incident light remains either

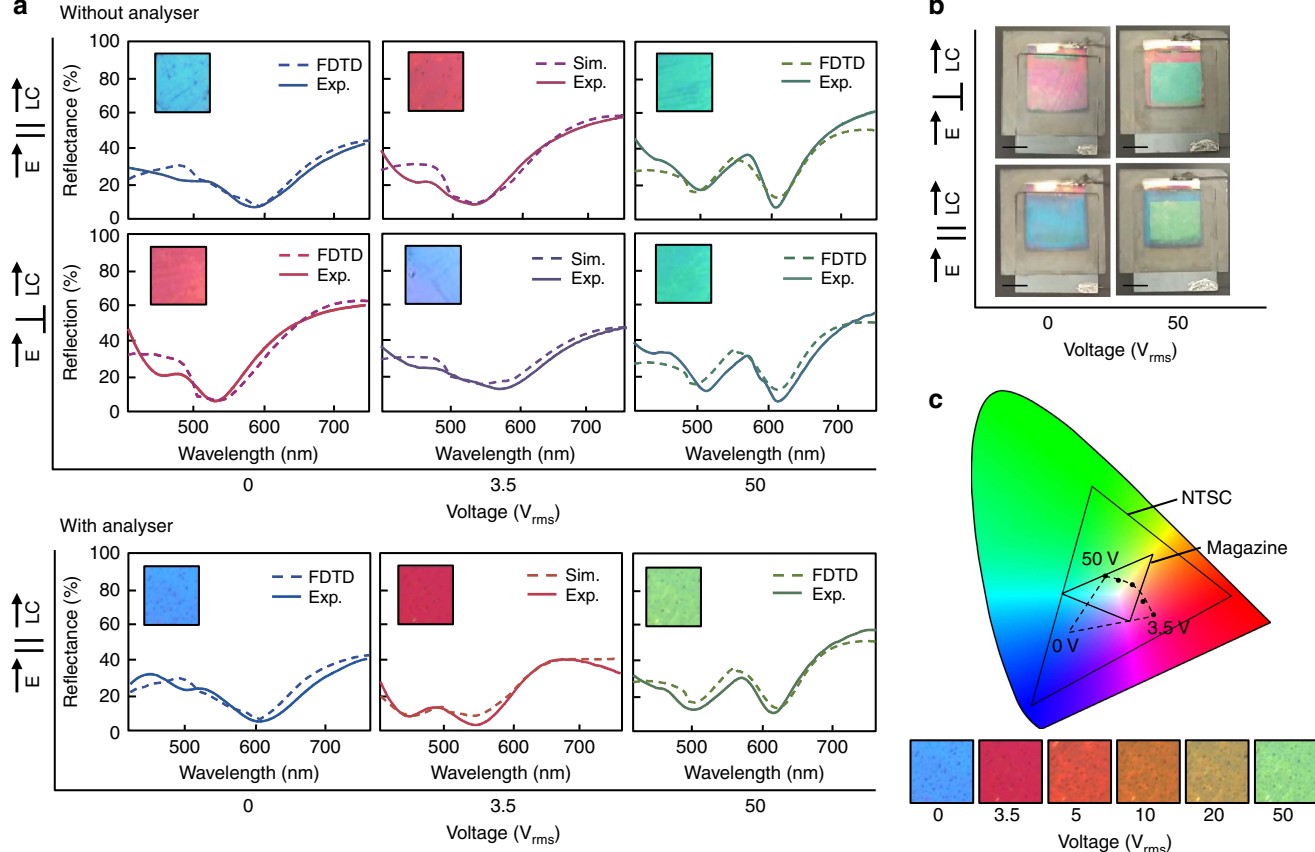

**Figure 4 | Full Red–Green–Blue colour and analyser dependence.** (**a**) Measured and simulated reflection spectra of the surface as a function of applied field and input polarization. The last row shows the influence of an added analyser parallel to the input polarizer. Insets show microscope images of the surface. Line colours are determined by the CIE colour matching functions. (**b**) Camera images of the device at selected polarizer orientations and voltage. Samples are 1 in × 1 in. (**c**) The CIE chromaticity diagram showing the colour space of the liquid crystal-plasmonic system and that of comparative standards. Below are microscope images of the surface at increasing applied fields. Scale bars, (**b**) 0.5 in.

parallel or perpendicular to the LC director as it excites the plasmonic surface. Within a low-voltage regime, where only the bulk LC changes orientation and the LC near the surface remains constant, the spectra is a linear combination of the two orthogonal basis spectra with a wavelength-dependent weighting, $\alpha$ and $\beta$. These weighting terms are given by the projection of the electric field exciting the nanostructure on the axes parallel and perpendicular to the LC director, respectively. Figure 4a shows experimentally obtained reflection spectra of the surface with a voltage of 3.5 V ($V_{\text{th}}$) where, for a given incident polarization the spectrum flips to the orthogonal state's colour. Using the Jones Matrix method to find $\alpha$ and $\beta$, the resulting simulated spectra closely match with experiment. Detailed steps for obtaining $\alpha$ and $\beta$, as well as their exact values for this case are shown in Supplementary Note 2.

As voltage is increased, the LC near the nanostructure and within the plasmonic fields of the surface begins to tilt. This increases the effective refractive index of the GCSP modes and continuously red shifts their resonant wavelengths. Once this occurs, the two-state approach used above is no longer valid, but also no longer needed to predict the device's end-state reflection spectra. At saturation voltage ($V_{\text{s}}$), the LC near the surface asymptotically approaches vertical alignment which results in the loss of the nanostructure's polarization dependence, as well as the bulk LC's birefringent properties. This is demonstrated in Fig. 4a where the voltage and polarization-dependent reflection spectra of the device is shown. Again, line colours are generated by overlap integration between the

measured spectra and the CIE colour matching functions and insets show optical microscope images of the surface. The two top rows of spectra are of the surface when excited with polarized light parallel and perpendicular to the LC alignment director, respectively, and light of all polarizations are collected. While this confirms that the colour is generated purely by the plasmonic surface, light from a practical device, in which the polarizer is laminated to the top, would pass through the polarizer a second time. The bottom-most row of spectra in Fig. 4a shows the influence of this second pass on the colour of the surface as an analyser is added in parallel with the polarizer. In the off-state, the spectra are invariant to the addition of an analyser as the LC is in the waveguiding regime as described in Fig. 3a. For this state, the experimental spectra match closely with the histogram-weighted FDTD spectra. At an applied voltage of 3.5 $V_{\text{rms}}$, the reflected colour flips due to the polarization rotation of incident light as it passes through the LC cell. The simulated spectra for this applied field is obtained by using the wavelength-dependent weighting factors, $\alpha$ and $\beta$, on the FDTD results at $V = 0$. The second pass through the polarizer is simulated by adding an analyser to the Jones calculus method. Here, we see a slight change in spectra with the addition of the analyser; however, the changes do not greatly impact the reflected spectra and colour. At saturation voltage (LC on state), the first-order plasmonic resonance shifts to 600 nm and a second-order resonance moves to 500 nm. This results in a green colour reflected from the surface which can be confirmed by FDTD with an anisotropic media where $n_z = n_e$ and $n_x = n_y = n_o$ of the LC. Here, we also see

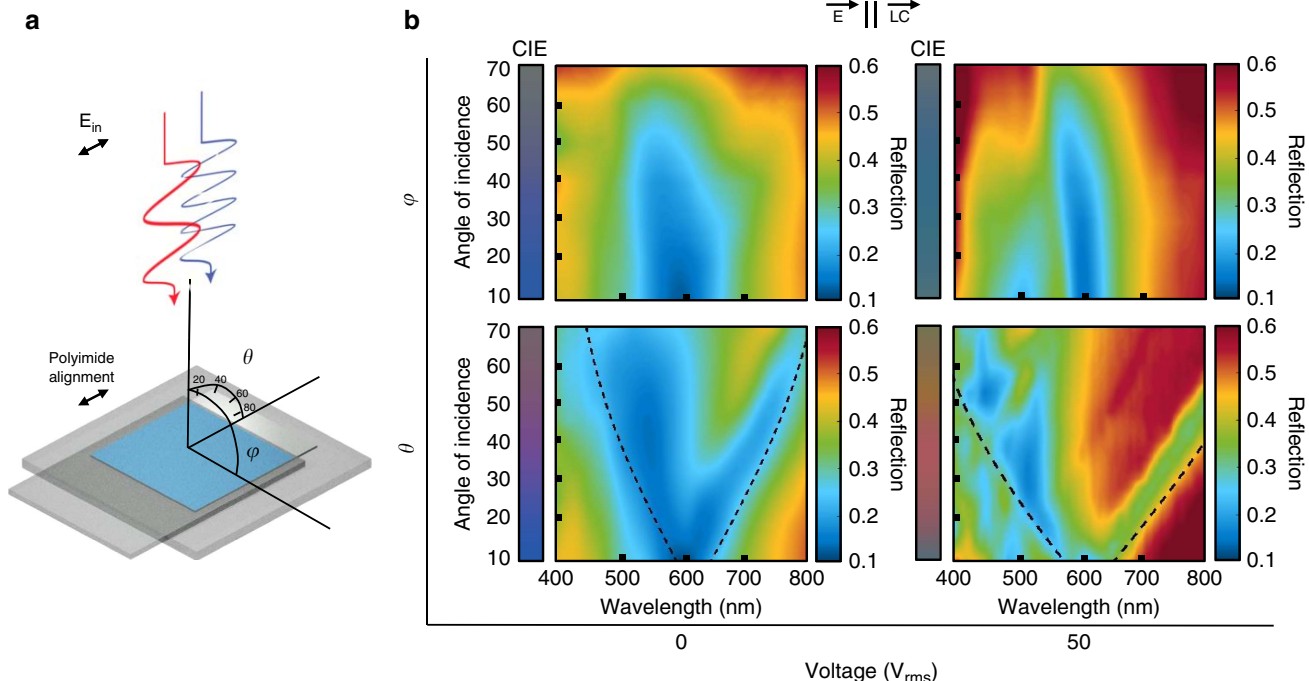

**Figure 5 | Angle dependence of the LC-plasmonic system.** (**a**) Schematic defining the polarization of incident light, liquid crystal alignment layer and angles of incidence, $\varphi$ and $\theta$. (**b**) Integrating sphere measurements of the device when the axis of rotation is parallel ($\varphi$, TE, s-polarized) and perpendicular ($\theta$, TM, p-polarized) to the incident polarized light. This is shown for both voltage extremes and where the LC alignment layer is parallel to the incident polarized light. CIE colours are obtained through the colour matching functions for each incident angle. Black lines depict the grating coupled surface plasmon (GCSP) relation when corrected for refraction through the top glass and LC interface.

that the effect of the analyser is minimal, which is consistent with the LC being nearly vertical throughout the cell. Figure 4b shows camera images of a large-area sample through a polarizing film for selected polarizations and voltages. Here, our devices are 1 in$^2$ in area and limited by the size and quality of our existing stamp; however, large area (36 in$^2$) and roll-to-roll nanoimprinting has been demonstrated that could allow scaling of the device to hand-held and notebook dimensions with high pattern fidelity. Another critical aspect of the resultant device is colour quality. Figure 4c shows the CIE chromaticity diagram and microscope images of the surface as a function of voltage when the polarizer, LC director and analyser are parallel. Black dots indicate the coordinates of the reflection spectra and a dotted black line shows the total area the surface could span if integrated in a colour-mixing scheme. Owing to the absorptive nature of the colour generating phenomenon, the colour gamut is less than that of the National Television System Committee (NTSC) standard but comparable in area to that of high-quality print magazine. A video of this colour-changing surface is shown in Supplementary Video 1.

**Angle dependence of the LC-plasmonic system.** To add to the fundamental study of the LC-plasmonic system, we perform angle resolved reflection measurements using an integrating sphere with a rotating centre mount (RTC-060-SF, Labsphere Inc.). A schematic is shown in Fig. 5a and depicts the case where incident light is polarized parallel to the LC director. We define two angles of incidence: $\varphi$, where the polarization of light is parallel to the axis of rotation (TE, s-polarized), and $\theta$, when the polarization of light is perpendicular to the axis of rotation (TM, p-polarized). For the incident angle $\varphi$, we see an expected angle independence of the resonance location from the experimental measurements in Fig. 5b. However, changes in excitation angle, $\theta$, produce a shift in resonant wavelength of the GCSP

modes according to the dispersion relation given above. Black lines depict this GCSP dispersion relation when corrected for refraction at the top glass and LC interfaces. To help understand the degree to which this dispersion impacts perceived colour, we take the spectra of Fig. 5b and calculate the CIE predicted colour—represented to the left of the spectral data. The results of these figures clearly show an area of improvement for a display utilizing this phenomenon. While an ideal display has little dependence on viewing position, this device has a relative angle invariance along one axis of rotation but a dependence in the other. However, by orienting the display along the preferred viewing angle, the display can be made almost angle invariant. Alternatively, the device could be used as a component in a projector or head mounted display in which the light source is at a fixed angle: taking advantage of such a device's sub-pixelless enhanced resolution while bypassing the issue of angle dependence altogether.

To improve upon the system's angle-independence, it is necessary to consider the plasmonic surface and the LC separately. As angular dispersion is a fundamental property of the GCSP mode, we believe the most straightforward way to improve the plasmonic absorber is to move to nanostructures which rely on alternative coupling mechanisms, such as metal–insulator–metal[15] and localized surface plasmon modes[32]. However, this may come at a cost in colour tuning capabilities as the overlap of plasmonic fields and LC, as well as near complete LC reorientation, may be insufficient. The LC contribution to the device's angle dependence is a well-studied area due to the demands of the LCD industry and there exist many strategies for improving it, such as birefringent compensation films[33], diffusors or lens arrays[34], and more drastically, changing which LC mode utilized[35].

**Active and passive addressing schemes.** Once a colour gamut is obtained, the system can be combined with various addressing

schemes. A critical advantage of the proposed LC-plasmonic system is its facile integration with pre-existing display technologies. To demonstrate this capability, we procured a conventional transmissive TN LCD panel (Adafruit, ID: 1680) and isolated the TFT array by removing the back light, polarizers, diffusers and ITO glass. The nanostructured aluminium surface is UV cured onto the TFT glass plate with 8.5 μm spacers and then filled with LCM1107. Figure 6a shows a microscope image of the resulting device (×10) in which the electrical components of the TFT can be clearly seen. Light passes through a polarizer, ITO windows and LC to reflect off the plasmonic surface and back out of the device. The white areas are due to reflection off the metal lines of the TFT. In the voltage-off state, the entire surface is either blue or red, depending on the polarization of incident light with respect to the top LC alignment director. By keeping the electronics of the TFT intact, the device is interfaced with a computer through a HDMI-to-TTL converter. Individual pixels are then controlled through images the computer outputs to the display. Figure 6b shows how the surface changes colour by applying a voltage to every third and fourth row of the TFT. Full images can be displayed as shown in Fig. 6c and video of text editing is demonstrated in Supplementary Video 2. While this shows the ease at which LC-plasmonic systems can be integrated with existing TFTs, the prototype device has several engineering challenges to overcome. The white reflection from the TFT metal lines, meant for use in transmissive displays, tends to wash out colour from the underlying plasmonic surface. A black matrix

needs to be superimposed on the TFT metal lines to mitigate this. Alternatively, fabrication of the plasmonic surface on the TFT glass itself will solve this unwanted reflection problem. Secondly, the off-the-shelf TFT drivers only source ~10 $V_{rms}$; not enough for the surface's transition to green for the present cell gap of 8.5 μm. TFT drivers capable of sourcing 15–30 V are needed in conjunction with a reduction in cell gap. The voltage needed for saturation is proportional to the thickness of the LC layer, but changes in this gap will alter the intermediate-voltage optical properties of the device. With proper engineering of these parameters, though, we believe the max operating voltage of the cell can be greatly reduced while maintaining its colour changing functionality. In absence of the required TFT in present academic setting, a passively addressed device is shown in Fig. 6d. Here, UV lithography is performed to macroscopically pattern the nanostructured surface, which is followed by a blanket deposition of aluminium. Treating this surface as the 'common', we pattern the top ITO glass to have individual control of each letter in 'UCF'. Furthermore, we utilize a UV photoalignment material[36] (PAAD-22, Beam Co.) instead of a physically rubbed polyimide to demonstrate how the off-state colours can be varied throughout a single device and with a single laminated polarizer. This azobenzene-based material, when exposed with linearly polarized UV light, aligns the LC homogeneously and perpendicularly to the polarization of exposure. Here, the alignment layer above the 'U' is exposed orthogonally with respect to the polarization of exposure above the 'CF'. The

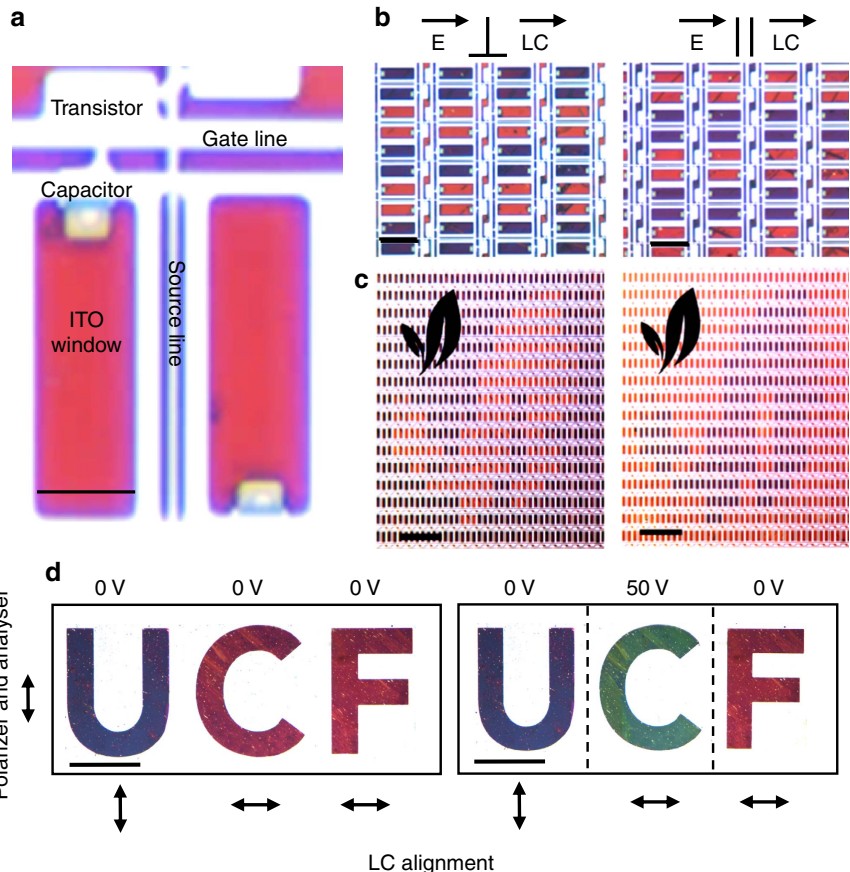

**Figure 6 | Passive and active addressing. (a)** A microscope image taken with a ×10 objective of the plasmonic surface integrated with a thin-film-transistor (TFT) array. Pixels can be controlled through a computer interface and **(b)** shows when every third and fourth row are turned on for the parallel and perpendicular incident light cases with respect to the top LC alignment layer. **(c)** Arbitrary images displayed from the device taken with a ×4 microscope objective. **(d)** A passively addressed device with UV lithographically patterned 'UCF' as a function of applied field, polarizer and analyser alignment and LC photoalignment director. Scale bars, **(a)** 30 μm, **(b)** 80 μm, **(c)** 0.72 mm, **(d)** 1.57 mm.

segmented top ITO/photoalignment substrate is then aligned with the 'UCF' plasmonic surface and the resulting cell is infiltrated with LCM1107. By changing the orientation of the LC alignment director with respect to the top polarizer, the 'U' and the 'CF' change between orthogonal colours. Lastly, we apply a field across the 'C' to obtain a green and demonstrate how the three RGB basis colours can be obtained from a single nanostructure/pixel.

Lastly, we consider two important aspects of the device: the fundamental pixel size and switching speed. The GCSP resonance requires multiple periods of the structure unit cell to effectively absorb light. We have found through simulation that ∼7 periods are needed for the resonance wavelength to be within 1 nm of the infinitely periodic case—corresponding to 2.1 µm pixel size for the 300 nm period structure[17]. While further studies are needed to quantify the impact of pixel size on colour quality, colour tuning and crosstalk, we believe this gives a conservative estimate for the minimum pixel size. The unoptimized switching characteristics of the device are shown in Supplementary Fig. 5 and show voltage-dependent rise and fall times. Full cycling times are somewhat invariant to voltage and in the 70 ms range, which equates to ∼14 Hz. While we believe this can be improved to standard display frame rates, it remains orders of magnitude faster than other colour-changing technologies, such as electrochemical[37] or translocation-based methods[38] which take seconds to tens of seconds to change.

## Discussion

While the proposed system demonstrates a full-colour tunable surface, complete commercial displays will also require mechanisms by which to control brightness. Black and grey states are essential to colour mixing processes and determine important properties such as contrast. While these states can theoretically be achieved in such an LC-plasmonic system, it remains a challenge to reconcile the different LC requirements needed for large colour tuning and brightness control—and even more so to control these properties independently within the same LC cell. We foresee critical advances in independent colour and brightness tuning utilizing combinations of surface and bulk LC phenomenon in conjunction with multiple types of electrode configurations. We believe the framework proposed herein can be translated to transmissive and/or transflective displays.

In summary, we have demonstrated a polarization-dependent LC-plasmonic system capable of producing the RGB colour basis set as a function of voltage and from a single surface. This is achieved using a single, continuous, aluminium nanostructure in conjunction with a high birefringence LC. By controlling the deposition conditions of the aluminium, a surface roughness-induced polarization dependence is realized in the presence of a surrounding anisotropic media. This phenomenon is then exploited through its integration with an LC cell which performs as a polarization rotator at low voltages while shifting the plasmonic resonance of the aluminium nanostructure at higher voltages. Large area nanoimprint lithography-based scalable fabrication of the nanostructure combined with its ready integration with LC technology can lead to a new class of LC-plasmonic devices.

## Methods

**Fabrication of nanostructured surfaces.** The nanostructured surface is fabricated through nanoimprint lithography (NIL) for rapid replication. A polymer (dimethylsiloxane) (PDMS; Dow Corning, Sylgard) stamp is cast from a master which has been made from beam lithography (EBL). A thin film of SU-8 2000.5 (MicroChem) was spun (500 r.p.m. for 5 s followed by 3,000 r.p.m. for 30 s) then prebaked at 95 °C for 1 min. This film is imprinted with the PDMS stamp (20 s) on the hotplate. The stamp and substrate are removed from the hotplate and allowed to cool (30 s). After stamp delamination, the substrate is UV cured (1 min) and post exposure baked (95 °C for 1 min).

**Electron beam deposition.** The 30 nm Al films are deposited using a Temescal (FC-2000) six-pocket electron beam evaporation system. For the 'smooth' film studied in Fig. 2c,e, the sample is mounted on a thermal electric cooler and brought to −20 °C. Evaporations are done at pressures of ∼6 × 10⁻⁶ T and deposition rates of ∼0.1 nm s⁻¹. For the 'rough' film used in Fig. 2d,f and throughout the rest of the paper, the samples are evaporated on at room temperature and starting pressures of ∼1 × 10⁻⁵ T. Before deposition, three edges of the sample were masked off. This greatly reduces the chance of a short circuit in the completed LC cell.

**LC cell formation.** The plasmonic LC cell is fabricated using commercially available twisted nematic LC cells (AWAT PPW, Poland). The commercial cells are heated to 200 °C and then split into two rubbed-polyimide ITO-coated glass slides with 8 µm silica spacers. A single slide is adhered to the plasmonic surface sample using NOA 81 with the polyimide alignment parallel to the nanostructure grating vector. Once UV cured, the LC-plasmonic cell is infiltrated with LC (LCM1107, LC Matter Corp.). The LC cells are driven with a 1 kHz AC sine wave to reduce ion migration. All reported voltages are RMS values unless stated otherwise.

**Optical measurements and images.** Reflection spectra are collected using a ×4, 0.07 numerical aperture objective on an optical microscope (Hyperion 1000) coupled to a Fourier transform infrared spectrometer (Vertex 80). Reflection spectra are normalized to an aluminium mirror with 96% reflectivity and a linear polarizer. Images are collected using the same optical microscope with an Infinity 2–5 camera. Defects due to stamp damage have been replaced by the nearest area in Fig. 5d with the GIMP software package.

**Finite difference time domain modelling.** Reflection spectra are calculated using experimental parameters for the printed 2D grating structures, with commercial FDTD software package (Lumerical FDTD, Lumerical Solutions Inc.). The profile for the electromagnetic simulations was obtained by fitting an analytical equation to SEMs of the nanostructured surface (Fig. 2c,d). Surface roughness profiles are generated in Matlab and imported into the FDTD simulations. The simulations of Fig. 2a,b are performed by changing the amplitude of a random variation about the top surface of the aluminium film. This has been done keeping the correlation length of the variation constant in the x and y directions at 15 nm. Periodic boundary conditions are used in the simulations, so the roughness within each unit cell is symmetric.

The wavelength-dependent refractive index of aluminium is taken from Palik and the anisotropic parameters of the LC layer are obtained using an effective anisotropic index model based on the orientation of LC obtained from FEM calculations.

**Data availability.** The data that support the findings of this study are available from the corresponding author on reasonable request.

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

## Acknowledgements

This work at University of Central Florida was supported by NSF grant No. ECCS-1509729.

## Author contributions

D.F. conceived the idea and D.F. and D.C. designed the experiments. D.F. and R.F. performed the experiments. D.F. analysed and simulated the data. D.C. and S.-T.W. contributed materials/analysis tools. D.F. and D.C. co-wrote the paper.

## Additional information

**Competing interests:** The authors declare no competing financial interests.

