## [Peer review file · Nature Communications]

Reviewers' comments:

Reviewer #1 (Remarks to the Author):

The authors develop a new plasmonic LC display mode. Their approach builds on a combination of anisotropy of plasmonic nano structures on surfaces and both bulk and surface effects within a liquid crystal slab placed atop of the plasmonic nanostructure. The team previously reported plasmonic LC display, also working in a reflection mode, but the stated breakthrough of the present manuscript is that the full color tuning can be achieved using applied voltage alone. From the standpoint of view of advancing plasmonic LC displays, this is certainly an important achievement, so, in general, I believe the manuscript can be appropriate for publication in Nature Communications. The manuscript is well written, the conclusions are supported by presented characterization results and modeling. The literature overview is reasonable although it certainly is missing other plasmonic LC approaches (see ACS Nano 9, 3097-3108; Nano Lett 14, 4071–4077) and mostly focuses on one approach of achieving tunable color, whereas other other approaches of LC-plasmonic voltage color tuning have not been discussed. Before I make my final recommendation, authors also should address the following points.

(1) While tuning pixel colors is important, display applications will require tuning brightness and other characteristics. What contrast ratios can be achieved? What is the quality of the dark state? I would like to see more characterization of these types of properties of the presented display mode.

(2) While reflective displays are of interest, majority of current applications require transmissive displays. Could the current approach be extended to achieve this? While the actual demonstration would be outside the scope of the present work, I would like to see an extension of discussion that addresses this important LC display application.

(3) The viewing angle characteristics seem to be such that the colors of displayed images will change as the customer will view the display from different angles. I would like to see a more extended discussion of how these issues can be mitigated and how they can be mitigated, say, by forming spatial patterns of plasmonic nanostructure anisotropy within individual sub pixels, etc.

Reviewer #2 (Remarks to the Author):

The paper entitled “Dynamically Tunable, Single Pixel Full-color Plasmonic Display” by D. Franklin et. al. and D. Chanda provides a simple yet versatile platform that demonstrates tunable reflective color filtering based on plasmonic nanostructures and Liquid crystals. The effect combines several physical mechanisms including grating coupled surface plasmon excitation; roughness and LC birefringence-induced polarization control of spectral filtering; and LC bulk reorientation-induced polarization rotation. After an in-depth demonstration of the physical mechanisms, the proposed design was successfully integrated into a commercial TFT array. The proposed platform could enable ultra-small pixel sizes in future display and imaging systems with further optimization of the LC-spacer layer thickness. I recommend the manuscript for publication, however I would suggest the following points to be addressed before final submission.

1. One of the key elements in this study is the polarization dependent color which is induced by the roughness control. The relevant section and Figure 2 present an in-depth investigation. However there are a few important points missing in the analysis: First, Fig. 2e and 2f give comparison of different polarizations at different roughness levels. In the experimental plots there are clearly two dips, apparently corresponding to first and second (diagonal) orders for the grating coupling. On the other hand, I don't see these dips are by a factor of $\sqrt{2}$ apart in wavelength, most likely because the second order peak (seen around 450nm) sees a higher effective index, which increases the resonance wavelength. Apart from that, simulations don't have a good overlap with the experimental dips (good match only with the first order mode). May be the simulations don't effectively reflect the anisotropy in the material index, which would lead second order peak appear at shorter wavelengths. Can authors provide a better analysis based on these arguments or based on their understanding for the discrepancy and the origin of these dips?

A second point is on the roughness control. Authors give a qualitative argument on the redshift seen at a particular polarization, based on the variation of the effective index seen by the plasmon resonance at different incident beam polarizations. Although the argument makes sense to some extent, a more detailed description for the nature of the resonant dip would give a clearer understanding. For instance the broadening in the resonances dips are mainly due to scattering loss, not due to having a distribution of grain diameters. The resonance is not a particle plasmon resonance, but rather a grating coupling resonance, so as the loss parameter is increased (with larger grain size), there will be less number of holes effectively interacting with each other, which will broaden the resonances (inducing a lower Q-factor). This could explain why the second order mode disappears in Fig.2f as well (the one at shorter wavelength), interaction dies as the interaction length increases – although virtually unchanged dips for the perpendicular polarization with such large grain sizes is unclear to me.

2. Although the descriptions for the polarization direction of the LC alignment and incident light sound technically correct, it is very hard to follow-up with, since there are several different cases. For instance, in the results section – line 97, LC alignment is described as a 45deg twist, gradually changing from one direction - aligned with the top rubbed layer, to a diagonal alignment to the nanowell grating vectors (which wasn't very clear by the way, until I saw Fig 3a). Then, in the “polarization dependent color” section authors assume a homogeneous LC alignment to isolate the bulk polarization rotation. This could be

acceptable for simulation purposes, however in the comparison of experimental and simulation results (Fig 2e-f), experiments will unavoidably have retardation and polarization rotation. I believe authors might be assuming it would make no difference, since the incident field would have a perpendicular or parallel polarization on the nanostructure surface for either case. On the other hand, for a reader trying to follow-up with the polarization status, it is very confusing. For instance, line 159 says incident light polarization is either parallel or perpendicular to the top rubbed alignment layer. Considering the fact that rubbed alignment layer is aligned with one of the nanowell grating vectors (as described in the results section - line 98) the reader will be lost because he/she was expecting a diagonal polarization – LC is supposed to be 45deg with nanowell periodicity, and polarization rotation has not yet been described until .

I don't have a particular suggestion to fix this, however one solution could be always using the same polarization description/assumptions etc. from the very beginning, and may be describing the polarization rotation at the beginning. Since at 0V bias, there is no bulk effect other than simple polarization rotation, your simulations in Fig.2 would be OK as well.

3. An important parameter to consider is pixel size. The particular structure under study has a great advantage to achieve submicron-size pixels. If possible, could the authors discuss the limitations of their design, for instance cross-talk of neighboring elements for an applied bias on one pixel, effect on the other? So how small could we go?

4. The expression in 294 is not accurate. For the first order mode (there is only one grating vector contribution) it works, however defining a general mode order using an integer m would be inaccurate. Second order mode would be the vectorial addition of the two grating vectors, which would give $\sqrt{2}$ instead of 2. I suggest keeping the simple equation and only using $m=1$ to define the first order, instead of using letter m , as a generalized equation... Otherwise, you could give a more complicated form of the equation as well.

A few editorial suggestions:

Line 56: periodicity should read periodicities

Figure 2e, colors in the plots are very similar, may be you could add labels next to the plots, one for 0deg, one for 90deg etc.

Reviewer #3 (Remarks to the Author):

This manuscript presents electrical tuning of full RGB colors by using liquid crystal-plasmonic systems, aiming at reflective color display applications. The authors have done extensive simulations and experiments in characterizing the system, and the results are interesting. This manuscript may be publishable in Nature Commn. I have the following questions/comments:

- (1) The electrical tuning of colors wider than full RGB spectrum can be done by using liquid crystals (LC) only, see paper: *Advanced Materials* 27, 3014-3018 (2015). Using plasmonic nanostructures for color tuning not only leads to high cost in manufacturing, but also limits reflection efficiency due to material loss (~20-40% in the visible range as shown in the paper). The plasmonic approach seems not very practical for applications. The authors need to explain what uniqueness the plasmonic structures can bring, and why this work and the results are important,.
- (2) It is not clear how FDTD simulations were set up for the films of different grain sizes. The grain sizes have a distribution, how that is taken into considerations? The lines 163-164 are not easy to understand.
- (3) The TechWiz simulations were described in the paper. It would be helpful if the authors can include some 3D director fields for the simulated structures, maybe as supplemental materials.
- (4) Minor issues. Some typos such as "superstrate". CIE needs to be spelled out at its first occurrence.

Report of Referee 1

Overall comment #1: “The authors develop a new plasmonic LC display mode. Their approach builds on a combination of anisotropy of plasmonic nano structures on surfaces and both bulk and surface effects within a liquid crystal slab placed atop of the plasmonic nanostructure. The team previously reported plasmonic LC display, also working in a reflection mode, but the stated breakthrough of the present manuscript is that the full color tuning can be achieved using applied voltage alone. From the standpoint of view of advancing plasmonic LC displays, this is certainly an important achievement, so, in general, I believe the manuscript can be appropriate for publication in Nature Communications.

Our response: We thank the reviewer for the positive recommendation and acknowledging the work as a significant advancement towards a working display based on the LC-plasmonic concept.

Overall Comment #2: “The manuscript is well written, the conclusions are supported by presented characterization results and modeling.”

Our response: We thank the reviewer for the positive comments on the write up and the extent we went to verify our experimental results.

Overall comment #3: “The literature overview is reasonable although it certainly is missing other plasmonic LC approaches (see ACS Nano 9, 3097-3108; Nano Lett 14, 4071–4077) and mostly focuses on one approach of achieving tunable color, whereas other other approaches of LC-plasmonic voltage color tuning have not been discussed. Before I make my final recommendation, authors also should address the following points.”

Our response: We agree that the literature overview can be further expanded and have attempted to make it more comprehensive by adding the suggested references [ACS Nano 9, 3097-3108; Nano Lett 14, 4071–4077] in the revised manuscript.

Our modification to the manuscript: Page 3, line 60: “*However, this and other reports of post-fabrication plasmonic tuning have yet to span an entire color basis set (RGB or CYM) with a single nanostructure¹⁸⁻²².*”

Comment #1: “While tuning pixel colors is important, display applications will require tuning brightness and other characteristics. What contrast ratios can be achieved? What is the quality of the dark state? I would like to see more characterization of these types of properties of the presented display mode.”

Our response: Complete commercial displays will certainly require properties as the reviewer suggests; dark and grey states. While these states could theoretically be achieved in such a LC-plasmonic system, it has proven difficult to reconcile the different LC requirements needed for large color tuning and brightness control - and harder still to control these properties independently within the same LC cell. In the present manuscript, we have decided to focus on the color changing aspects of such a system and to extend it to cover the whole RGB color space.

We intend to address the dark/grey states in our near future publication. To highlight this point, we have added a new paragraph to the revised manuscript.

Our modification to the manuscript: Page 18, line 406: *“While the proposed system demonstrates a full-color tunable surface, complete commercial displays will also require mechanisms by which to control brightness. Black and grey states are essential to color mixing processes and determine important properties such as contrast. While these states can theoretically be achieved in such a LC-plasmonic system, it remains a challenge to reconcile the different LC requirements needed for large color tuning and brightness control - and even more so to control these properties independently within the same LC cell. We foresee critical advances in independent color and brightness tuning utilizing combinations of surface and bulk LC phenomenon in conjunction with multiple types of electrode configurations in near-future.”*

Comment #2: “While reflective displays are of interest, majority of current applications require transmissive displays. Could the current approach be extended to achieve this? While the actual demonstration would be outside the scope of the present work, I would like to see an extension of discussion that addresses this important LC display application.”

Our response: We agree with the reviewer’s assessment and believe that this approach can be extended to widely used transmissive displays as well. We have added a new paragraph in the revised manuscript to highlight this aspect.

Our modification to the manuscript: Page 19, line 414: *“We believe the framework proposed herein can be translated to transmissive and/or transflective displays.”*

Comment #3: “The viewing angle characteristics seem to be such that the colors of displayed images will change as the customer will view the display from different angles. I would like to see a more extended discussion of how these issues can be mitigated and how they can be mitigated, say, by forming spatial patterns of plasmonic nanostructure anisotropy within individual sub pixels, etc.”

Our response: We have shown in Figure 5 that for one axis of rotation the system remains angle-independent, but as the reviewer correctly states, changes with rotation of the other axis. To help understand the degree to which this problem exists and its impact on perceived color, we have taken the measured spectra of Figure 5b and calculated the CIE predicted color. This data has been added to Figure 5-b and shows that the degree of color change will depend greatly on the specific voltage and polarizer configuration of the device. However, to put this in context, early transmissive displays exhibited similar issues such as color inversion and we believe this proposed platform can be improved as well. To do so, it is necessary to separately analyze the angle dependence of the plasmonic surface and that of the LC.

The plasmonic nanostructure studied herein absorbs light through a grating coupled surface plasmon resonance which is inherently angle dependent. The 2D nature of the grating then allows the color of the surface to be angle dependent for p-polarized light while independent for s-polarized light. As this is a fundamental property of this coupling mode, we believe the easiest way to improve angle dependence is to move to a nanostructure with angle-independent coupling modes, as demonstrated in [Sci Rep. 3, 1194 (2013)] or [Adv. Optical Mater. 4, 1981-

1986 (2016)]. However, this may come at a cost in color tuning capabilities. On the other hand, the LC contribution to the device's angle dependence is more involved as it will change as a function of voltage. The method for determining angle dependence is well documented and there exist many strategies for improving it, such as birefringent compensation films, microprisms or lens arrays, and more drastically, changing which LC mode utilized. We are actively working to improve the angular response of the combined plasmonic-LC system based on an independent architecture which is beyond the scope of this present manuscript and will be part of a near future publication. We have added a new paragraph to the revised manuscript to elaborate this point.

Our modification to the manuscript: Page 15, line 326: *“To help understand the degree to which this dispersion impacts perceived color, we take the spectra of Fig 5 (b) and calculate the CIE predicted color – represented to the left of the spectral data. The results of these figures clearly show an area of improvement for a display utilizing this phenomenon. While an ideal display has little dependence on viewing position, this device has an angle invariance along one axis of rotation but a dependence in the other. However, by orienting the display along the preferred viewing angle, the display can be made almost angle invariant. Alternatively, the device could be used as a component in a projector or head mounted display in which the light source is at a fixed angle: taking advantage of such a device’s sub-pixeless enhanced resolution while bypassing the issue of angle dependence altogether.*

To improve upon the system’s angle-independence, it’s necessary to consider the plasmonic surface and the LC separately. As angular dispersion is a fundamental property of the GCSP mode, we believe the most straightforward way to improve the plasmonic absorber is to move to nanostructures which rely on alternative coupling mechanisms, such as metal-insulator-metal¹⁵ and localized surface plasmon modes³². However, this may come at a cost in color tuning capabilities as the overlap of plasmonic fields and LC, as well as near complete LC reorientation, may be insufficient. The LC contribution to the device’s angle dependence is a well-studied area due to the demands of the LCD industry and there exist many strategies for improving it, such as birefringent compensation films, diffusors or lens arrays³⁴, and more drastically, changing which LC mode utilized³⁵.”

Our modification to the Figure 5:
Added CIE color bars corresponding to the angle dependent reflection spectra in Fig. 5.

Figure 5 | Angle Dependence of the LC-Plasmonic System. (a) Schematic defining the polarization of incident light, liquid crystal alignment layer and angles of incidence, ϕ and θ . (b) Integrating sphere measurements of the device when the axis of rotation is parallel (ϕ , TE, s-polarized) and perpendicular (θ , TM, p-polarized) to the incident polarized light. This is shown for both voltage extremes and where the LC alignment layer is parallel to the incident polarized light. CIE colors are obtained through the color matching functions for each incident angle. Black lines depict the grating coupled surface plasmon (GCSPP) relation when corrected for refraction through the top glass and LC interface.

Report of Referee 2

Overall comment #1: “The paper entitled “Dynamically Tunable, Single Pixel Full-color Plasmonic Display” by D. Franklin et. al. and D. Chanda provides a simple yet versatile platform that demonstrates tunable reflective color filtering based on plasmonic nanostructures and Liquid crystals. The effect combines several physical mechanisms including grating coupled surface plasmon excitation; roughness and LC birefringence-induced polarization control of spectral filtering; and LC bulk reorientation-induced polarization rotation. After an in-depth demonstration of the physical mechanisms, the proposed design was successfully integrated into a commercial TFT array. The proposed platform could enable ultra-small pixel sizes in future display and imaging systems with further optimization of the LC-spacer layer thickness. I recommend the manuscript for publication, however I would suggest the following points to be addressed before final submission.”

Our response: We thank the reviewer for the positive recommendation. We hope to address the reviewer’s concerns to the best of our abilities in the revised manuscript.

Comment #1, Part 1: “One of the key elements in this study is the polarization dependent color which is induced by the roughness control. The relevant section and Figure 2 present an in-depth investigation. However, there are a few important points missing in the analysis: First, Fig. 2e and 2f give comparison of different polarizations at different roughness levels. In the experimental plots there are clearly two dips, apparently corresponding to first and second (diagonal) orders for the grating coupling. On the other hand, I don’t see these dips are by a factor of $\sqrt{2}$ apart in wavelength, most likely because the second order peak (seen around 450nm) sees a higher effective index, which increases the resonance wavelength. Apart from that, simulations don’t have a good overlap with the experimental dips (good match only with the first order mode). May be the simulations don’t effectively reflect the anisotropy in the material index, which would lead second order peak appear at shorter wavelengths. Can authors provide a better analysis based on these arguments or based on their understanding for the discrepancy and the origin of these dips?

Our response: The origin of the spectral dips are grating coupled surface plasmon resonances

defined by the analytical equation, $\frac{2\pi}{\lambda} = \frac{P}{\sqrt{i^2 + j^2}} \sqrt{\frac{\epsilon_{Al}\epsilon_{LC}}{\epsilon_{Al} + \epsilon_{LC}}}$, where P is the period of the

grating, i and j are mode orders, and ϵ_{Al} , ϵ_{LC} are the permittivity for aluminum and surrounding LC, respectively. The permittivity of aluminum depends greatly on grain size, surface roughness (rms height variation and correlation length) and oxide content. While these are difficult to obtain analytically, we can determine them for our samples by measuring the spectra when exposed to an isotropic media (like the portion of the sample where NOA UV glue is used to adhere the two substrates). We believe discrepancies arise from the ϵ_{LC} term, which in this equation is assumed to be isotropic and uniform. In reality, the effective ϵ_{LC} experienced by the plasmonic fields is an overlap integral of the LC director tensor with the plasmonic fields. However, these entities are difficult to quantify as the LC director tensor will influence/distort

the plasmonic fields and the LC director will depend greatly on variations in the surface. Due to these complexities, we have taken a first order approach and assumed a constant anisotropic effective index, ϵ_{LC} , in the simulations of the surface, independent of surface roughness. By varying this term, we can obtain accurate fitting of the first order mode, but as the reviewer stated, the higher order mode experiences a different LC refractive index as the plasmonic field profile of the mode is ultimately different and the real LC director is not that of a uniformly aligned anisotropic crystal.

Taking these higher order effects into account is of interest but would require much further work. For the purpose and scope of this work, however, we believe the data shows that the first order method utilized herein is adequate to predict/confirm the color perceived by the surface. We have added several sentences to the revised manuscript to emphasize this point.

Our modification to the manuscript: Page 6, line 116: “The color of the surface originates from absorptive GCSP resonances characterized by²⁷ $\vec{k}_{GCSP} = \frac{2\pi}{\lambda_0} \sqrt{\frac{\epsilon_{AL}\epsilon_{LC}}{\epsilon_{LC} + \epsilon_{LC}}} \vec{u} + i \frac{2\pi}{P_i} \vec{i} + j \frac{2\pi}{P_j} \vec{j}$.”

Here, \vec{k}_{GCSP} is the GCSP wavevector, λ_0 the incident wavelength, ϵ_{LC} and ϵ_{AL} the dielectric constants of the LC and aluminum, respectively, \vec{u} is the unit vector of the surface plasmon before bragg scattering, i and j are the mode orders, and P_i and P_j are the structure periodicities along the grating vectors \vec{i} and \vec{j} .”

Our modification to the manuscript: Page 7, line 141: “*In reality, the LC director aligns to the contours of the aluminum surface, resulting in a complex LC director tensor dependent on the local topography of the surface. However, to isolate the effects of rms surface roughness on the GCSP resonance alone, we choose to simplify this LC layer and leave it constant for the purposes of the numerical demonstration in Fig. 2 (a-b).*”

Our modification to the manuscript: Page 8, line 156: “*We explain this phenomenon with the following argument: the ϵ_{LC} term of the above equation can be thought of as an effective index experienced by the plasmonic mode. For isotropic and uniform surrounding media, this term reduces to the dielectric constant of the medium. However, for anisotropic media, this term becomes an overlap integral between the plasmonic field tensor and the surrounding media’s dielectric tensor.*”

Our modification to the manuscript: Page 10, line 205: “*We believe discrepancies between the experimental and numerical spectra of Fig. 2(e-f) occur for the higher order mode due to the approximation of the LC as a uniform and anisotropic crystal.*”

Comment #1, Part 2: “A second point is on the roughness control. Authors give a qualitative argument on the redshift seen at a particular polarization, based on the variation of the effective index seen by the plasmon resonance at different incident beam polarizations. Although the argument makes sense to some extent, a more detailed description for the nature of the resonant dip would give a clearer understanding. For instance the broadening in the resonances dips are mainly due to scattering loss, not due to having a distribution of grain diameters. The resonance is not a particle plasmon resonance, but rather a grating coupling resonance, so as the loss parameter is increased (with larger grain size), there will be less number of holes effectively

interacting with each other, which will broaden the resonances (inducing a lower Q-factor). This could explain why the second order mode disappears in Fig.2f as well (the one at shorter wavelength), interaction dies as the interaction length increases – although virtually unchanged dips for the perpendicular polarization with such large grain sizes is unclear to me.

Our response: We agree with the reviewer that an increase in scattering loss induces broadening in plasmonic resonances. While this may explain why the resonances of Figure 2 f are wider than Figure 2 e (comparing for like polarizations), as the reviewer pointed out, this doesn't explain why the higher order resonance of Figure 2 f has disappeared for the 0° case but remain in the 90° case. We believe this could be explained, though, in the variations of the LC director about the surface. The LC directs itself homogeneously along the surface, contouring its tilt along the bumps and grains of the surface. Plasmons excited with light polarized along this direction (n_e) will experience these variations in refractive index, and the increased scattering these variations induce. Plasmons excited with the perpendicular polarization (along n_o) would uniformly experience the small axis of the LC even as the LC director aligns to the contours of the aluminum in the orthogonal direction.

Our modification to the manuscript: Page 6, line 125: *“This red-shift is accompanied by resonance broadening as an increase in scattering also decreases the interaction length of the plasmon resonance - inducing a lower Q-factor.”*

Comment #2, Part 1: “Although the descriptions for the polarization direction of the LC alignment and incident light sound technically correct, it is very hard to follow-up with, since there are several different cases. For instance, in the results section – line 97, LC alignment is described as a 45deg twist, gradually changing from one direction - aligned with the top rubbed layer, to a diagonal alignment to the nanowell grating vectors (which wasn't very clear by the way, until I saw Fig 3a). Then, in the “polarization dependent color” section authors assume a homogeneous LC alignment to isolate the bulk polarization rotation. This could be acceptable for simulation purposes, however in the comparison of experimental and simulation results (Fig 2e-f), experiments will unavoidably have retardation and polarization rotation. I believe authors might be assuming it would make no difference, since the incident field would have a perpendicular or parallel polarization on the nanostructure surface for either case. On the other hand, for a reader trying to follow-up with the polarization status, it is very confusing. For instance, line 159 says incident light polarization is either parallel or perpendicular to the top rubbed alignment layer. Considering the fact that rubbed alignment layer is aligned with one of the nanowell grating vectors (as described in the results section- line 98) the reader will be lost because he/she was expecting a diagonal polarization – LC is supposed to be 45deg with nanowell periodicity, and polarization rotation has not yet been described until . I don't have a particular suggestion to fix this, however one solution could be always using the same polarization description/assumptions etc. from the very beginning, and may be describing the polarization rotation at the beginning. Since at 0V bias, there is no bulk effect other than simple polarization rotation, your simulations in Fig.2 would be OK as well.”

Our response: Firstly, we would like to thank the reviewer for this thorough and thoughtful comment.

One of the main contentions in the formation of the manuscript has been the order in which we discuss the polarization dependent color effects and that of the bulk LC rotation. We have opted to discuss them as we have because the two polarization dependent color states of the device (red and blue) were needed to explain the color obtained at the intermediate voltage (which is a superposition of these two states based on the polarization rotation that occurs at this voltage). However, we agree that the polarization analysis can be difficult to follow. Hence, we have made a number of changes to the revised manuscript with the purpose of alleviating this.

Also, we believe the comparison of the experimental data and simulations of Figure 2 e-f are valid as this figure solely considers the 0V state in which the polarization of light stays either completely parallel or perpendicular to the LC. We have attempted to emphasize this by adding new sentences to the revised manuscript.

Our modification to the manuscript: Page 9, line 182: *“We show below that in the 0 V state, light polarized parallel or perpendicular to the top rubbed polyimide layer is guided along the twist of the LC as it propagates to the plasmonic surface. Therefore, the experimental polarization excitation (which is with respect to the top polyimide) is equivalent to that of the simulation (which is with respect to the LC director) and we label them 0° and 90°, respectively.”*

Comment #3: “An important parameter to consider is pixel size. The particular structure under study has a great advantage to achieve submicron-size pixels. If possible, could the authors discuss the limitations of their design, for instance cross-talk of neighboring elements for an applied bias on one pixel, effect on the other? So how small could we go?”

Our response: The grating coupled surface plasmon resonance requires multiple periods of the structure to effectively absorb light and we have attached below a supplementary figure from our previous publication (Franklin et.al., Nature Communications, Vol. 6, pp. 7337, June 2015.) to help describe this. Obtained through FDTD simulations, it shows how the form of the reflection spectra changes as we change the number of nanostructure periods. We assume the LC is in a complete vertical state for the simulations. As the number of periods increases, the resonance shifts, narrows and becomes more pronounced. We found ~7 periods are needed for the resonance to be within 1 nm of the infinitely periodic case, corresponding to 2.1 μm pixel size for the 300 nm structure. Even with seven periods, though, a broadening is seen compared to the infinitely periodic case. As the pixel size decreases we would expect to see a shift in reflected color, until it washes out as the resonance becomes weaker and weaker.

Our modification to the manuscript: Page 18, line 392: *“Lastly, we consider two important aspects of the device: the fundamental pixel size and switching speed. The GCSP resonance requires multiple periods of the structure unit cell to effectively absorb light. We have found through simulation that ~7 periods are needed for the resonance wavelength to be within 1 nm of the infinitely periodic case – corresponding to 2.1 μm pixel size for the 300 nm period structure¹⁷. While further studies are needed to quantify the impact of pixel size on color quality, color tuning and crosstalk, we believe this gives a conservative estimate for the minimum pixel size.”*

Supplementary Fig. 8 from D. Franklin, et al., “Polarization Independent, Actively Tunable Color Generation on Imprinted Plasmonic Surfaces”, Nature Communications, Vol. 6, pp. 7337, June 2015.

Supplementary Figure 8 | Pixel size dependence. FDTD simulations for a structure of period 300 nm and 100 nm relief depth. (a), FDTD reflection spectra of structures with varying numbers of periods. (b) The first order resonant wavelength as a function of surfaces with varying numbers of periods. We find that the resonant wavelength location approaches that of the infinitely periodic structure within 1 nm for 9 structure periods.

Comment #4: “The expression in 294 is not accurate. For the first order mode (there is only one grating vector contribution) it works, however defining a general mode order using an integer m would be inaccurate. Second order mode would be the vectorial addition of the two grating vectors, which would give $\sqrt{2}$ instead of 2. I suggest keeping the simple equation and only using $m=1$ to define the first order, instead of using letter m , as a generalized equation... Otherwise, you could give a more complicated form of the equation as well.”

Our response: We thank the reviewer for catching this mistake and we have made changes to rectify it.

Our modification to the manuscript: Page 15, line 323: “However, changes in excitation angle produce a shift in resonant wavelength of the GCSP modes according to³¹ $\vec{k}_{GCSP} = \frac{2\pi}{\lambda_0} \sqrt{\frac{\epsilon_{Al}\epsilon_{LC}}{\epsilon_{LC} + \epsilon_{LC}}} \vec{u} + i \frac{2\pi}{P_i} \vec{i} + j \frac{2\pi}{P_j} \vec{j}$. Here, \vec{k}_{GCSP} is the GCSP wavevector, λ_0 the incident wavelength, ϵ_{LC} and ϵ_{Al} the dielectric constants of the LC and aluminum, respectively, \vec{u} is the unit vector of the surface plasmon before bragg scattering, i and j are the mode orders, and P_i and P_j are the structure periodicities along the grating vectors \vec{i} and \vec{j} .”

Editorial Suggestion #1: “Line 56: periodicitys should read periodicities”

Our response: We thank the editor for pointing out this typo.

Our modification to the manuscript: Page 3, line 58: “In our previous work, we demonstrated that the range of plasmonic color tuning could span the visible spectrum by using nanostructures of several periodicities in conjunction with a high birefringent liquid crystal (LC)¹⁷.”

Editorial Suggestion #2: “Figure 2e, colors in the plots are very similar, may be you could add labels next to the plots, one for 0deg, one for 90deg etc.”

Our response: We have added labels to Figure 2 e to help identify the corresponding spectra.

Our modification to Figure 2:

Converted Max Roughness to RMS Roughness on the y-axis of (a-b).

Added labels to (e)

Report of Referee 3

Overall comments #1: “This manuscript presents electrical tuning of full RGB colors by using liquid crystal-plasmonic systems, aiming at reflective color display applications. The authors have done extensive simulations and experiments in characterizing the system, and the results are interesting. This manuscript may be publishable in Nature Commn. I have the following questions/comments:”

Our response: We thank the reviewer for the positive evaluation of the work. We tried to address review comments in the revised manuscript.

Comment #1: “The electrical tuning of colors wider than full RGB spectrum can be done by using liquid crystals (LC) only, see paper: *Advanced Materials* 27, 3014-3018 (2015). Using plasmonic nanostructures for color tuning not only leads to high cost in manufacturing, but also limits reflection efficiency due to material loss (~20-40% in the visible range as shown in the paper). The plasmonic approach seems not very practical for applications. The authors need to explain what uniqueness the plasmonic structures can bring, and why this work and the results are important.”

Our response: The paper the reviewer cites describes a cholesteric LC cell that reflects circularly polarized light and is tunable from UV to NIR wavelengths. We agree that the work is very impressive as it achieves an incredible range of tuning with a narrow band of light (high contrast color). However, the system is not without downsides; the cell requires between 40 V (IR state) and 200 V (UV state) to operate, and has large switching times of 800 ms (UV to red) to 2 min (0V to red state) of switching time. The LC-plasmonic system, here, requires less than 50 V to produce RGB and does so in approximately 70 ms. As the plasmonic color of the surface is determined by interfacial phenomenon, we believe both of these parameters can be drastically improved by reducing cell gap ($V \propto d$) while still maintaining the polarization rotation of the bulk LC due to its high birefringence. Both systems have comparable reflection amplitudes (they depict up to 41% for visible light, we demonstrate up to 35%).

A distinct advantage of plasmonic color over other systems is its scale. Plasmonic systems have demonstrated diffraction limited color and can produce the smallest color generating display elements physically possible. Along with the ability to control phase and polarization, this could lead to incredibly small pixels useful for next generation projection or 3D displays. For these reasons, we find the possibility of hybrid LC-plasmonic displays and devices worth exploring.

We agree with the reviewer and have elaborated on the unique properties of plasmonic structures in the revised manuscript - which we believe will help the manuscript connect with a larger audience.

Our modification to the manuscript: Page 3, line 43: “*Structural color arising from plasmonic nanomaterials and surfaces has received ever increasing attention¹⁻¹². These nanostructures have demonstrated diffraction limited color through the subwavelength confinement of light and can produce the smallest color generating elements physically possible. Along with the ability to*

control phase and polarization, these metallic nanostructures could lead to very small pixels useful for next generation projection or 3D displays.”

Comment #2: “It is not clear how FDTD simulations were set up for the films of different grain sizes. The grain sizes have a distribution, how that is taken into considerations? The lines 163-164 are not easy to understand.”

Our response: The FDTD simulations of Figure 2 are performed by changing the amplitude of a uniform random variation about the top surface of the aluminum film. This has been done keeping the correlation length of the variation constant in the x and y directions. Periodic boundary conditions are used in the simulations, so the roughness within each quadrant the unit cell is symmetric.

To obtain a simulation based comparison of the experimental spectra in Figure 2 e-f, we begin by finding the grain size distribution of the two films, Figure 2 c-d. Image processing software is used to obtain a histogram of the variation in grain diameters for each film respectively. We then use the Gaussian fit of the histogram (red curve in Figure 2 c-d) in a weighted average of the FDTD spectra in Figure 2 a-b. We do this to more accurately represent the surface roughness distribution of the experimental films, which differ from the uniform random distribution of the simulations.

We agree that the inter-use of terms to quantify the surface, such as grain diameter, rms surface roughness and correlation length is not clear and may create confusion. We have attempted to clarify this by adding details about the relationship between these terms in the revised manuscript and modifying the labels of Figure 2 (a-b).

Our modification to the manuscript: Page 6, line 131: *“To demonstrate the impact of rms surface roughness on the color reflected from the nanostructure, we perform FDTD simulations of the periodic nanowell array by changing the amplitude of a uniform random height variation about the top surface of the aluminum film while maintaining a constant lateral correlation length of 15 nm. We then use the resulting spectra to predict a reflected color through the 1932 International Commission on Illumination (CIE) color space and color matching functions. Further details about the surface generation and simulations can be found in the Methods section.”*

Our modification to the manuscript: Page 9, line 189: *“As grain diameter and rms surface roughness are proportional, we use the histograms of the respective aluminum surfaces as a weighted average of the FDTD spectra of Fig. 2(a-b) to more accurately represent the surface roughness distribution of the experimental films, which differ from the uniform random distribution of the simulations.”*

Our modification to the manuscript: Page 21, line 466: *“Surface roughness profiles are generated in Matlab and imported into the FDTD simulations. The simulations of Figure 2 (a-b) are performed by changing the amplitude of a uniform random variation about the top surface of the aluminum film. This has been done keeping the correlation length of the variation constant in the x and y directions at 15 nm. Periodic boundary conditions are used in the simulations, so the roughness within each the unit cell is symmetric.”*

Our modification to the manuscript: Page 5, line 92: “We use this to create a relatively rough aluminum surface (~30 nm grain diameter) which induces a polarization dependence on the GSCP resonance in the presence of an anisotropic media, whereas it was shown in our previous work that a smooth surface (~10 nm grain diameter) results in near polarization independence¹⁷”.

Our modification to Figure 2:

Converted Max Roughness to RMS Roughness on the y-axis of (a-b).
 Added labels to (e)

Comment #3: “The TechWiz simulations were described in the paper. It would be helpful if the authors can include some 3D director fields for the simulated structures, maybe as supplemental materials. “

Our response: We agree with the reviewer and examples of the 3D director fields along with the twist and tilt of the LC as a function of cell location and voltage are already shown in the supplementary information, Fig. S3.

Our modification to the manuscript: Page 10, line 220: *“A detailed description of the process and examples of the 3D director fields along with the twist and tilt of the LC as a function of cell location and voltage are shown in the Supplementary Information.”*

Comment #4: “Minor issues. Some typos such as "superstrate". CIE needs to be spelled out at its first occurrence.”

Our response: We thank the reviewer for pointing these out and have made the appropriate changes in the revised manuscript. We intended the use of the term “superstrate” as to “substrate” to indicate it being at the top of the device stack.

Our modification to the manuscript: Page 7, line 135: *“We then use the resulting spectra to predict a reflected color through the 1932 International Commission on Illumination (CIE) color space and color matching functions.”*

REVIEWERS' COMMENTS:

Reviewer #1 (Remarks to the Author):

authors appropriately accounted for my suggestions and remarks - I recommend publication

Reviewer #2 (Remarks to the Author):

In the manuscript entitled "Dynamically Tunable, Single Pixel Full-color Plasmonic Display", the authors convincingly addressed all the points that have been raised, and made adequate changes in the manuscript accordingly. I recommend the manuscript for publication.

Just as an editorial suggestion: in the updated manuscript, Page 15, line 334, there is a typo and an unclear statement: "...source is at a fixed angle: taking advantage of such a device's sub-pixeless enhanced resolution ...".

Reviewer #3 (Remarks to the Author):

The authors have addressed the questions carefully, and I recommend it for publication in Nature Communications.

Report of Referee 1

Comment #1: “authors appropriately accounted for my suggestions and remarks - I recommend publication”

Our response: We thank the reviewer for his/her evaluation of our response.

Report of Referee 2

Overall comment #1: “In the manuscript entitled "Dynamically Tunable, Single Pixel Full-color Plasmonic Display", the authors convincingly addressed all the points that have been raised, and made adequate changes in the manuscript accordingly. I recommend the manuscript for publication.”

Our response: We thank the reviewer for his/her evaluation of our response.

Comment #1: “Just as an editorial suggestion: in the updated manuscript, Page 15, line 334, there is a typo and an unclear statement: "...source is at a fixed angle: taking advantage of such a device’s sub-pixeless enhanced resolution ...”

Our response: We believe the reviewer is referring to the word “sub-pixeless” and we have corrected it within the manuscript.

Our modification to the manuscript: Page 15, Line 335: “sub-pixelless”

Report of Referee 3

Overall comments #1: “The authors have addressed the questions carefully, and I recommend it for publication in Nature Communications.”

Our response: We thank the reviewer for his/her evaluation of our response.